# Community Detection in Shipping Network Based on AIS data

Xin Zhang
Collaborative Innovation Center
for Transport Studies
Dalian Maritime University
Dalian, China
xinzhang23@dlmu.edu.cn

Yi Zuo
Navigation College
Dalian Maritime University
Dalian, China
zuo@dlmu.edu.cn

Junhao Jiang
Navigation College
Dalian Maritime University
Dalian, China
jiangjunhao@dlmu.edu.cn

Peng Jia
Collaborative Innovation Center
for Transport Studies
Dalian Maritime University
Dalian, China
jiapeng@dlmu.edu.cn

*Abstract—In response to the burgeoning shipping industry and its impact on congested waterways, port authorities face heightened regulatory pressures and navigation safety challenges. To aid these authorities, we propose a community discovery method for analyzing vessel traffic networks using AIS data. This method identifies stationary and turning points in vessel trajectories, applies feature-based clustering, extracts node attributes, and utilizes frequent trajectory pattern recognition to map waterway route networks. Through empirical analysis of cargo ship data from Ningbo-Zhoushan Port, our approach effectively identifies critical areas such as stationary and turning zones, aligning closely with real-world scenarios. This method offers port authorities timely insights into high-risk areas for safe navigation and supports optimized route planning, demonstrating its accuracy, effectiveness, and practicality.*

*Keywords—automatic Identification System (AIS); shipping network; community discovery; network analysis*

## I. INTRODUCTION

Existing studies predominantly focus on maritime shipping routes between ports, treating ports as nodes and shipping routes as edges, but often overlook the local shipping trajectories along these routes and lack clear insights into maritime route networks within water bodies. Addressing these gaps, this study utilizes AIS big data to extract critical nodes from shipping trajectories, constructing networks of stationary and turning areas within local water bodies. By analyzing vessel trajectory characteristics, the study aims to identify high-risk areas using community detection methods. This approach helps maritime authorities focus on high-risk areas promptly, ensuring safe navigation for vessels and facilitating route planning[2].

In recent years, with the widespread application of Automatic Identification System (AIS) data, the analysis of vessel trajectory data has become a significant research direction in maritime and shipping management. The focus of research has gradually expanded from basic data preprocessing to encompass various aspects such as complex pattern recognition and community detection.Due to the typically large amount of noise and incomplete information in AIS data, data preprocessing is a critical step in the analysis. Researchers have proposed various methods to clean and reconstruct trajectories to ensure data accuracy and completeness. Hu and Zhang discussed handling missing data, filtering noise points, and reconstructing continuous vessel trajectories [3]. Pallotta et al. utilized Hidden Markov Models (HMM) for trajectory reconstruction and prediction [4].Li, Liu, and Chen proposed methods based on density clustering (such as DBSCAN), segmentation techniques, and graphical analysis to identify these key points and extract useful attribute information [5]. Zhang et al. used density-based clustering algorithms to segment vessel trajectories and identify critical behavioral patterns [6].Wan, Li, and Zhang established complex maritime traffic network models using methods from graph theory and network science to analyze connections and network characteristics between different regions [7]. Kaluza et al. studied the network properties of global maritime traffic, revealing its complex topological structure [8].Zhao and Shi utilized the Louvain algorithm, Girvan-Newman algorithm, and other community detection techniques to successfully identify critical maritime zones and navigation nodes, providing insights to enhance navigation safety [9]. Xu et al. researched density-based clustering methods for community detection to identify high-density navigation areas [10].

This paper focuses on categorizing vessel navigation states into straight sailing, stationary, and turning, with emphasis on stationary and turning as critical features. It constructs a cargo ship feature point network model from preprocessed AIS data, comprising five modules, and performs community detection and exploration analysis. In AIS data preprocessing, we utilize AIS data to filter out vessel data in the Ningbo-Zhoushan Archipelago, and perform cleaning and reconstruction to obtain high-quality trajectory point data. In trajectory feature point identification and attribute extraction, we identify stationary points and turning points from trajectory data, extract node attributes accordingly, cluster them to form feature regions, and generate region attributes from node attributes. In extraction of network structure edges, we use frequent trajectory patterns [11] to identify and assess connections between feature regions. In network topology construction, we utilize the Louvain algorithm to establish a maritime route network topology graph based on marked representative points of feature regions and

Identify applicable funding agency here. If none, delete this text box.

extracted edges. In community detection, we return to map analysis.

## II. CONSTRUCTION OF SHIPPING ROUTE NETWORK

### A. Preprocessing AIS Data

The original AIS data, containing both static and dynamic information, is processed by decoding and reconstructing it according to IEC and ITU specifications. Relevant AIS data within the study area is extracted using vessel traffic service queries based on MMSI identifiers. Trajectories are generated by sequentially connecting points along each vessel's path based on timestamps, forming polyline graphs. To ensure data accuracy, noise points are manually filtered out, resulting in clean AIS trajectory data for analysis.

### B. Identification and Extraction of Trajectory Keypoints

#### 1) Identification of Stationary Points

When a ship is stationary, its speed is 0 or close to 0 Points in a trajectory where a ship's speed is less than 0.5 are defined as stationary points. The set of stationary points ($P_S$) is identified using density-based clustering (DBSCAN) algorithm, which efficiently detects stationary areas and calculates attributes such as the time interval between a stationary point and the next normal sailing point on the same trajectory.

The duration of stays in these aforementioned areas varies. The time interval is defined as an attribute of stationary points. The $P_S$ set, represented by centroids of stationary areas, is obtained using DBSCAN clustering. These centroids ($P_{SD}$) effectively represent the stationary situation of vessels in these areas. The coordinates of stationary area centroids are the latitude and longitude of the DBSCAN circle centers, and the attribute of each centroid is the average value of stationary point attributes during DBSCAN clustering. Analyzing the average stationary time in an area can effectively analyze and determine the region and status of vessels during their voyages.

#### 2) Identification and Extraction of Turning Points:

By calculating the rate of change in heading at each time point, which is the difference in heading between consecutive points on the same track, significant turning points ($P_t$) can be identified. These turning points are crucial for analyzing specific turning behaviors of the vessel.

The set of turning points Pt is then processed using density-based spatial clustering of applications with noise (DBSCAN) algorithm to extract clusters, swiftly identifying turning regions. From these clusters, representative points are obtained, denoted as $P_{tD}$, which serve as centroids for the turning regions. The coordinates of these representative points correspond to the latitude and longitude of the DBSCAN cluster centers, and their attributes represent the average properties of the turning points within each cluster during the DBSCAN clustering process. Analyzing the average rate of heading change within a region provides valuable insights for maritime management, shipping companies, and other stakeholders, enabling the identification of hazardous areas and collision avoidance points.

### C. Extraction of Network Structure Edges

The article constructs separate networks for stationary feature zones and turning feature zones, termed as the stationary feature zone network and turning feature zone network respectively. During the extraction of network structure edges, connectivity is only assessed between stationary feature zones, disregarding connections between stationary and turning feature zones. The same principle applies to turning feature zones.

#### 1) Construction of Adjacency Matrix

Due to the large number of trajectory points and the complexity of trajectories between feature areas, only frequent trajectories are selected as the research objects [12]. Each feature area is considered a node in the network topology structure. A threshold value (f=15) is set. The number of trajectories containing points in two feature areas is calculated, and it is determined whether they are connected in adjacent time within the same trajectory. If connected, a trajectory between the two feature areas is recorded. By traversing numerous trajectory points, the number of trajectories (s) between two feature areas is computed. If ( s>f ), it indicates that the connection between the two feature areas, nodes (m) and (n), is frequent. Thus, the two feature areas can be considered connected [11], as shown in Figure 1.To determine frequent trajectory patterns between each pair of feature areas (nodes), an adjacency matrix can be obtained.

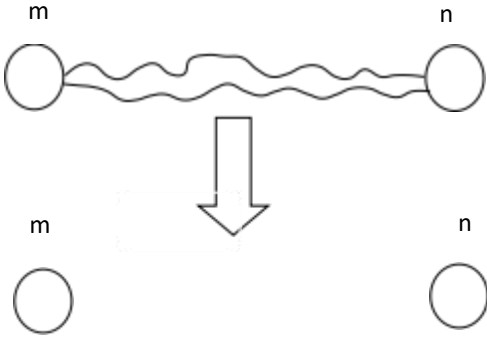

(a)Do not establish edges between infrequent feature zones

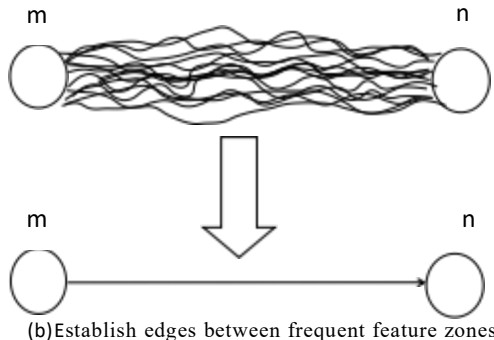

(b)Establish edges between frequent feature zones

Fig. 1.  Frequent trajectory patterns

#### 2) The Construction of Weight Matrix

In this study, stationary and turning feature areas are treated as network nodes with specific attributes. Stationary feature areas are characterized by their average dwell time, while turning feature areas are defined by their average heading change rate. These attributes not only capture the duration of stay and movement patterns within each area but also serve as crucial metrics for subsequent network analysis.

To effectively incorporate these node attributes into the network topology, we will use attribute similarity as the weight for node connections [13]. For each pair of nodes, we compute the difference between their attribute values and then normalize these differences to standardize them, enabling comparisons across varying value ranges. These normalized values are utilized to build a weight matrix, where each element denotes the similarity between nodes and acts as the edge weight in the network.

Specifically, for nodes (m) and (n) with attribute values $A_m$ and $A_n$ respectively, he attribute difference between the two nodes is given by $(\left(\left|A_m - A_n\right|\right))$. To normalize this difference , we will use Equation (1):

$$W_{mn} = 1 - \frac{|A_m - A_n|}{\max|A_m - A_n|}$$

（1）

Where $W_{mn}$ represents the weight between nod ⁽¹⁾ d node n, and $\max\left(\left|A_m - A_n\right|\right)$ denotes the maximum difference among all pairs of node attributes. This normalization ensures that each element in the weight matrix ( W ) ranges between 0 and 1, where 1 indicates high similarity and 0 indicates no similarity.

This attribute-based weighted network model not only preserves the original node attribute information but also visually demonstrates relationships between different feature areas through the network structure. This method provides a new dimension for exploring potential interactions among feature areas, aiding in a deeper understanding of behavioral patterns in complex systems.

### D. Network Topology Construction

After obtaining the adjacency matrix and the weight matrix, in order to efficiently process large-scale networks and identify optimal community structures quickly, especially in complex feature networks involving numerous nodes and edges, particularly when seeking local optimal community structures, we choose the Louvain algorithm for community detection. Louvain algorithm flowchart as shown in Figure 2.

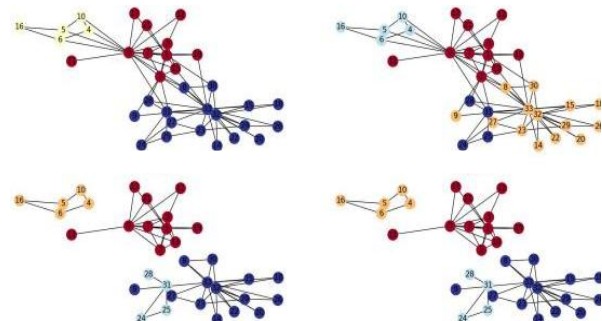

Fig. 2. Louvain algorithm iteration process

### E. Community Detection

Identifying densely connected subgraphs in complex feature networks reveals inherent community structures, aiding in understanding network organization, functionality, and properties. Mapping these communities back to original marine data involves analyzing their features and attributes. Integrating community and marine attribute analyses deepens insights into spatial distribution in marine environments, navigation system structures, and their stability [14].

### III. CASE STUDY

Ningbo-Zhoushan Port is an important port on the eastern coast of mainland China, comprising Ningbo Port and Zhoushan Port. Located in Zhejiang Province, it is one of China's largest and among the busiest ports globally.

### A. Data Preparation

This study utilized AIS data from Ningbo-Zhoushan Port, covering the geographical coordinates (121°43'.999E to 122°36'.899E, 29°38'.670N to 30°22'.728N) during January 1 to June 30, 2018, as shown in Figure 3. Analyzing cargo vessel trajectories in this maritime area is crucial for enhancing port efficiency, safety, and generating economic and environmental benefits. This research significantly contributes to the regional and global maritime economy.

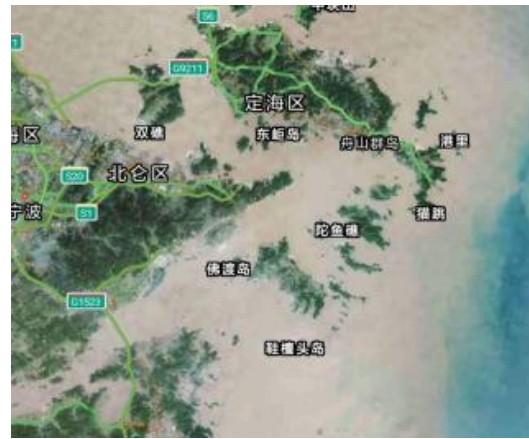

Fig. 3. The geographical location of Ningbo-Zhoushan Port

Therefore, this study decided to analyze the trajectory points of cargo ships. To minimize interference from other vessels, only data classified as cargo ships were selected from the raw data. After data cleaning and noise reduction, the trajectory points of each MMSI number were observed. Unreasonable points were removed, and the resulting data was visualized using Python, as shown in scatter plot Figure 4.

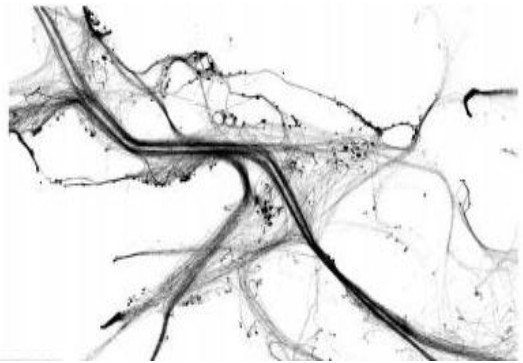

Fig. 4. Scatter plot of cargo ship trajectories

## B. Extraction of Network Nodes

### 1) Extraction of Points for Characteristic Regions

According to the rules for identifying characteristic points, using Python, we filtered out 115,848 stationary points and 21,360 turning points from the original dataset. Visualizations of the stationary points are shown in Figure 5a, and the turning points in Figure 5b.

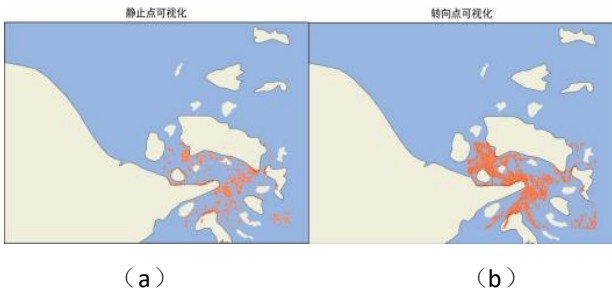

Fig. 5. Visualization of feature points

For clustering the characteristic points, we employed the DBSCAN algorithm. After iterative adjustments and referencing silhouette coefficients, we found that setting the parameter $\varepsilon$ to 200 meters and $P_{min}$ to 25 yielded satisfactory results for clustering stationary feature area centroids, resulting in 396 representative points, as depicted in Figure 6a. Setting $\varepsilon$ to 150 meters and $P_{min}$ to 30 achieved improved clustering effectiveness for turning feature area centroids, yielding a total of 374 representative points, as shown in Figure 6b.

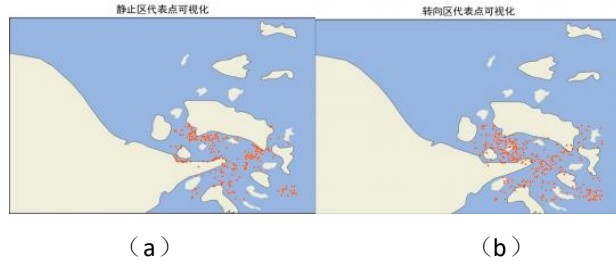

Fig. 6. Visualization of centroids of feature area points

### 2) Extraction of Attributes from Feature Areas

In trajectory data processing, Unix timestamps are first decoded into standard date-time formats for analysis. Stationary points, where there's minimal positional change over time, and turning points, indicating significant directional changes, are identified by analyzing trajectory characteristics.

For stationary points, the duration to the nearest non-stationary point is calculated initially. Clustering algorithms like DBSCAN group these points based on geographic proximity, forming distinct stationary feature areas. The average duration of points within each cluster represents its centroid. Similarly, turning points' initial attribute is the rate of heading change to the nearest non-turning point. Clustering based on geographic location identifies multiple turning feature areas, with each cluster's centroid computed from the average heading change rate.

This approach efficiently extracts attribute features from trajectory data, delineating feature areas precisely. It supports further trajectory analysis and network construction using centroids of 395 stationary feature areas (orange in Figure 7) and 373 turning feature areas (blue dots).

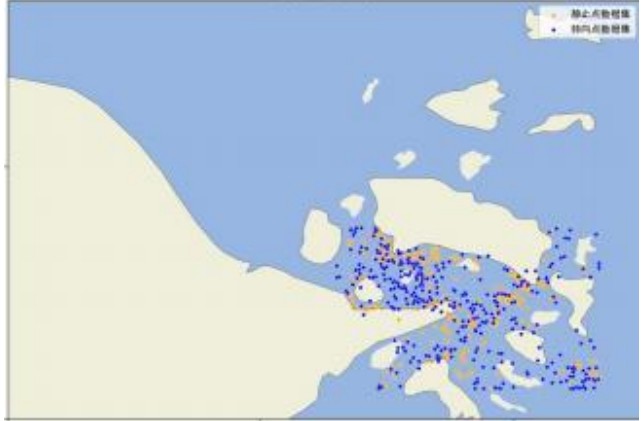

Fig. 7. Visualization of Centroids of Feature Regions

## C. Network Topology Analysis

In this study, we applied the Louvain algorithm for community detection on static and dynamic feature networks. This method identified nodes sharing similar attributes, facilitating deeper analysis of network structures. Community detection and visualization techniques were crucial for understanding connectivity among feature regions and enhancing trajectory data analysis.

For the static feature network, we utilized Gephi and Python to visualize the network and its communities. Figures 10 and 11 display the visual outcomes of this analysis. Additionally, these tools enabled computation of key network metrics such as node degree, modularity, network diameter, and average path length.

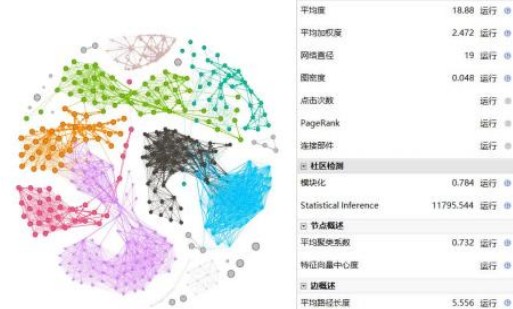

Fig. 8. tatic feature network and its metrics

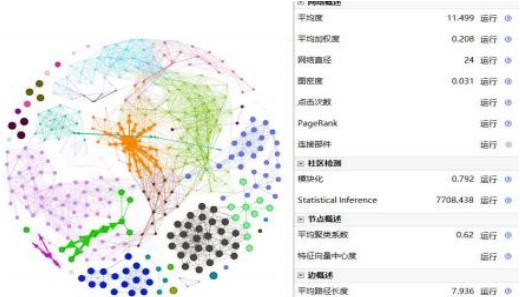

Fig. 9. Turning feature network and its metrics

## D. Community Detection

To gain a more intuitive understanding of sub-communities, nodes belonging to different sub-communities are depicted in the Ningbo-Zhoushan Port waters. Nodes within the same community share similar attributes and features. The visualizations of sub-communities in the static feature network and turning feature network are shown on the map in Figures 12 and 13, respectively.

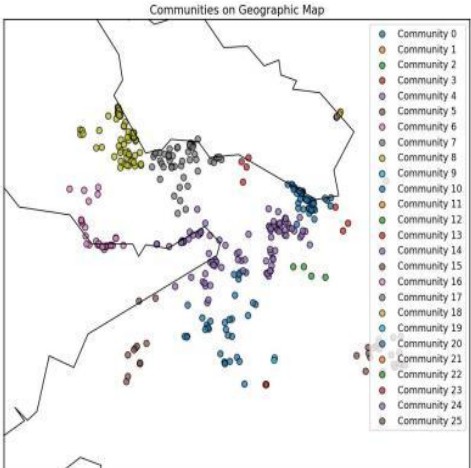

Fig. 10. Sub-communities of the static feature network

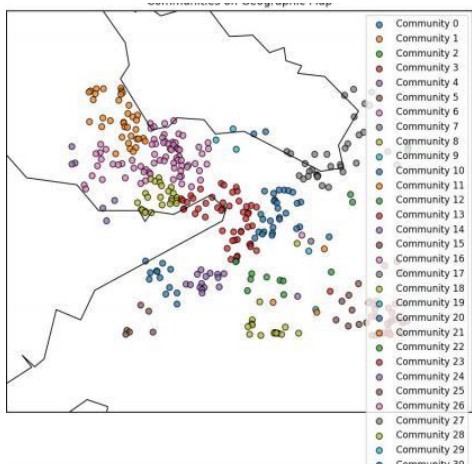

Fig. 11. sub-communities of the turning feature network

In the static feature network, we observed the network divided into 26 sub-communities, while the turning feature network was segmented into a total of 36 sub-communities. In the static feature network, we observed a division into 26 sub-communities, while the turning feature network was divided into a total of 36 sub-communities. However, some of these sub-communities might be overly simplistic or lack sufficient structural characteristics. Therefore, we opted not to visualize them further for additional analysis. Table 1 presents the static sub-communities studied.

Table 2 calculates the numerical values of degree centrality, closeness centrality, and betweenness centrality [18] for each node within the sub-communities of the static feature network, identifying nodes with the highest centrality.

| Sub-community labels | Visualization of sub-communities |
|---|---|
| Community 0 | 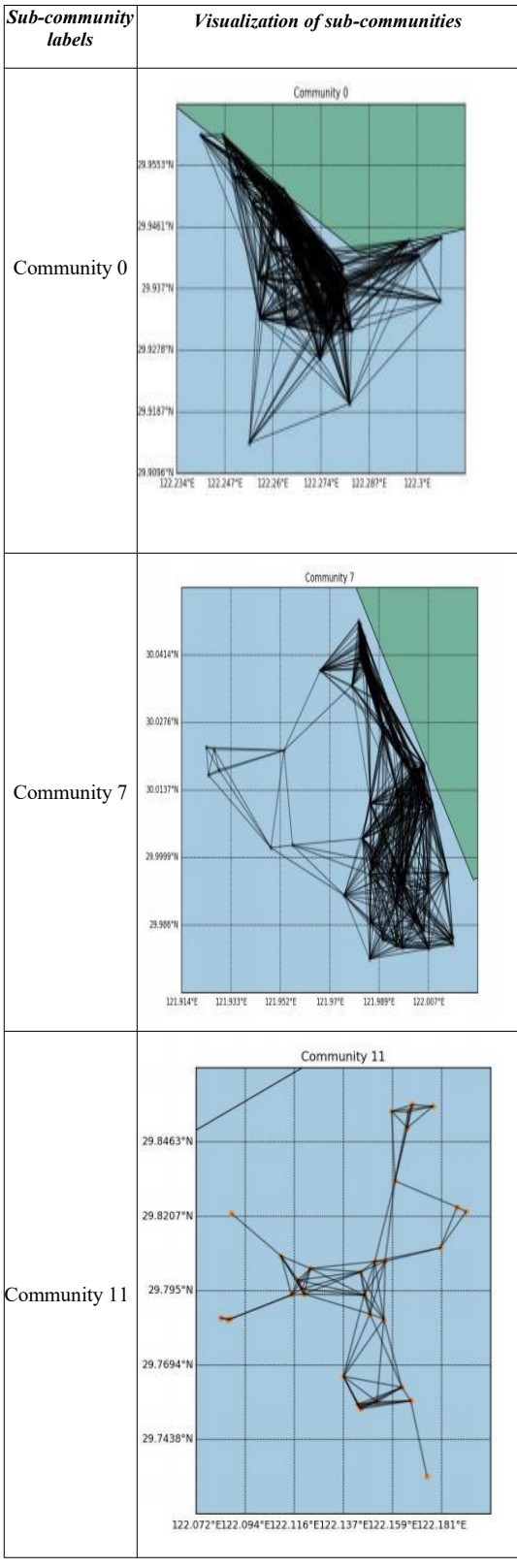 |
| Community 7 | |
| Community 11 | |

Average degree, Network diameter and Average path length analysis metrics are computed using Gephi, as shown in Table 3. We separately remove the nodes with the highest

centrality in each sub-community—highest degree centrality in Table 4, highest closeness centrality in Table 5, and highest betweenness centrality in Table 6—and analyze their impact on the network's average degree, network diameter, and average path length.

TABLE II.    STATIC FEATURE NETWORK SUB-COMMUNITIES

| Sub-community labels | Community 0 | Community 7 | Community 11 |
|---|---|---|---|
| Degree centrality highest node | 56 | 249 | 226 |
| Degree centrality | 0.144 | 0.108 | 0.031 |
| Closeness centrality highest node | 280 | 13 | 226 |
| Closeness centrality | 0.148 | 33 | 0.078 |
| Betweenness centrality highest node | 280 | 214 | 67 |
| Betweenness centrality | 0.007 | 0.044 | 0.028 |

Average degree, Network diameter and Average path length analysis metrics are computed using Gephi, as shown in Table 3. We separately remove the nodes with the highest centrality in each sub-community—highest degree centrality in Table 4, highest closeness centrality in Table 5, and highest betweenness centrality in Table 6—and analyze their impact on the network's average degree, network diameter, and average path length.

TABLE III.    ANALYSIS OF STATIC NETWORK METRICS

| Average Degree | 18.880 |
|---|---|
| Network Diameter | 19.000 |
| Average Path Length | 5.556 |

TABLE IV.    ANALYSIS OF NETWORK METRICS AFTER REMOVING NODES WITH HIGHEST DEGREE CENTRALITY

| Sub-community labels | Community 0 | Community 7 | Community 11 |
|---|---|---|---|
| Average Degree | 18.867 | 18.713 | 18.867 |
| Network Diameter | 19.000 | 19.000 | 19.000 |
| Average Path Length | 5.655 | 5.598 | 5.547 |

TABLE V.    ANALYSIS OF NETWORK METRICS AFTER REMOVING NODES WITH HIGHEST CLOSENESS CENTRALITY

| Sub-community labels | Community 0 | Community 7 | Community 11 |
|---|---|---|---|
| Average Degree | 18.641 | 18.738 | 18.798 |
| Network Diameter | 19.000 | 19.000 | 19.000 |
| Average Path Length | 5.572 | 5.565 | 5.748 |

TABLE VI.    ANALYSIS OF NETWORK METRICS AFTER REMOVING NODES WITH HIGHEST BETWEENNESS CENTRALITY

| Sub-community labels | Community 0 | Community 7 | Community 11 |
|---|---|---|---|
| Average Degree | 18.641 | 18.754 | 18.892 |
| Network Diameter | 19.000 | 19.000 | 20.000 |
| Average Path Length | 5.572 | 5.561 | 5.650 |

After removing the node with the highest degree centrality, the average degree across the entire network decreased by 0.064 on average, with a change rate of 3.407‰. The average change in network diameter was 0, and the average path length increased by 0.044 on average, with a change rate of 7.920‰.After removing the node closest to the highest degree centrality, the average degree across the entire network decreased by 0.154 on average, with a change rate of 8.174‰. The average change in network diameter was 0, and the average path length increased by 0.072 on average, with a change rate of 13.020‰.After removing the node with the highest betweenness centrality, the average degree across the entire network decreased by 0.118 on average, with a change rate of 6.232‰. The average network diameter increased by 0.333 on average, with a change rate of 17.544‰, and the average path length increased by 0.038 on average, with a change rate of 6.900‰.

Through the study, it was found that in the directional feature network derived using the Louvain algorithm, removing the node closest to the highest degree centrality had the greatest impact on the average degree of the network. Removing the node with the highest betweenness centrality had the greatest impact on the average network diameter, while removing the node closest to the highest degree centrality had the greatest impact on the average path length of the network. Removing any critical node had the largest impact on the average path length of the network, with a relatively smaller effect on the network diameter.

Through research, it has been found that within the static feature network identified using the Louvain algorithm, the removal of nodes with the highest closeness centrality has the greatest impact on average degree, while the removal of nodes with the highest betweenness centrality has the greatest impact on average network diameter. Removing nodes with the highest closeness centrality has the greatest impact on average path length. Removing any key node has the greatest impact on average path length but a relatively smaller impact on network diameter. Based on this conclusion, the following recommendations are proposed: For ports, enhancing berthing infrastructure is crucial to boost capacity and operational efficiency, meeting growing cargo demands and reducing vessel waiting times. At waterway convergence points, managing navigation routes and traffic flow while mitigating collision risks is essential for maritime safety. Anchorage planning and utilization must focus on efficiency through strategic layout and facility improvements, along with stringent

management standards to ensure safe operations. Tailored infrastructure development and management strategies tailored to static key nodes are vital for enhancing maritime transportation efficiency and safety across the network.

The turning sub-communities studied are listed in Table 7.

TABLE VII.    DYNAMIC FEATURE NETWORK SUB-COMMUNITIES

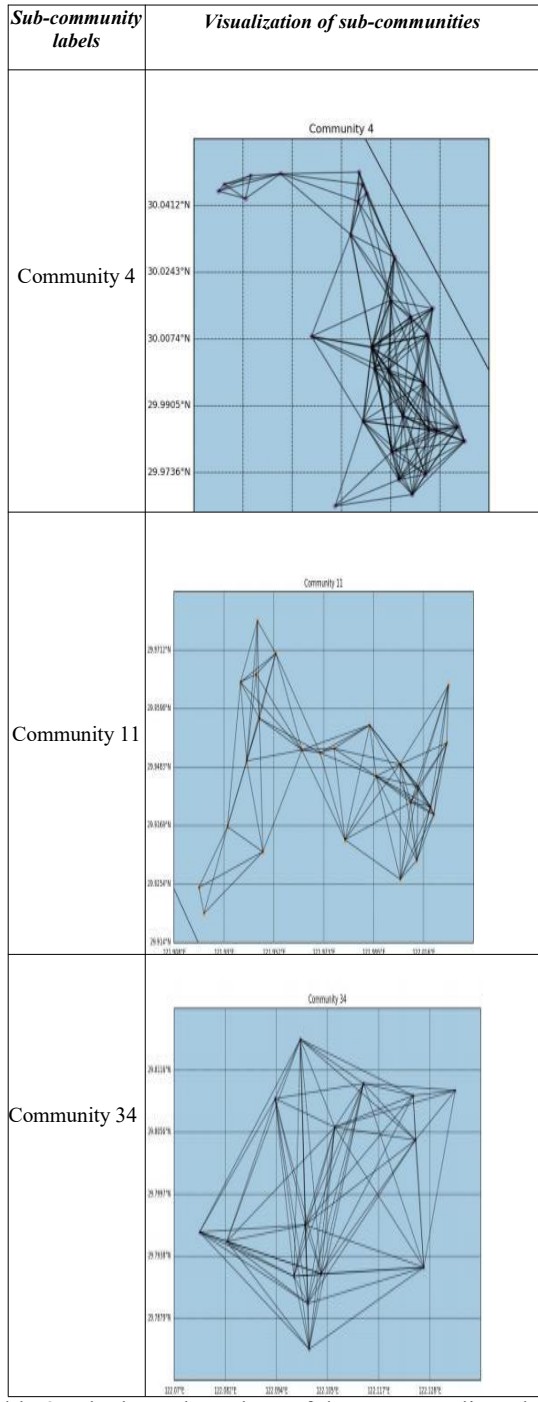

| Sub-community labels | Visualization of sub-communities |
|---|---|
| Community 4 | |
| Community 11 | |
| Community 34 | |

Table 8 calculates the values of degree centrality, closeness centrality, and betweenness centrality for each node within sub-communities of the static feature network, identifying the node labels with the highest centrality.

Table 9 presents the calculations of average degree, network diameter, and average path length using Gephi. We proceed to analyze the impact on these network metrics by sequentially removing the most central nodes based on highest centrality within each sub-community, as shown in Table 10 for highest degree centrality nodes, Table 11 for highest closeness centrality nodes, and Table 12 for highest betweenness centrality nodes.

TABLE VIII.    DYNAMIC FEATURE NETWORK SUB-COMMUNITIES

| Sub-community labels | Community 4 | Community 11 | Community 34 |
|---|---|---|---|
| Degree centrality highest node | 50 | 92 | 287 |
| Degree  centrality | 0.06 | 0.059 | 0.035 |
| Closeness centrality highest node | 54 | 133 | 284 |
| Closeness centrality | 0.113 | 0.121 | 0.109 |
| Betweenness centrality highest node | 15 | 102 | 285 |
| Betweenness centrality | 0.023 | 0.029 | 0.063 |

TABLE IX.    DYNAMIC FEATURE NETWORK SUB-COMMUNITIES

| Average Degree | 11.499 |
|---|---|
| Network Diameter | 24.000 |
| Average Path Length | 7.936 |

TABLE X.    TABLE IV. ANALYSIS OF NETWORK METRICS AFTER REMOVING NODES WITH HIGHEST DEGREE CENTRALITY

| Sub-community labels | Community 4 | Community 11 | Community 34 |
|---|---|---|---|
| Average Degree | 11.411 | 11.411 | 11.459 |
| Network Diameter | 24.000 | 24.000 | 24.000 |
| Average Path Length | 7.940 | 7.944 | 7.936 |

TABLE XI.    TABLE V. ANALYSIS OF NETWORK METRICS AFTER REMOVING NODES WITH HIGHEST CLOSENESS CENTRALITY

| Sub-community labels | Community 4 | Community 11 | Community 34 |
|---|---|---|---|
| Average Degree | 11.427 | 11.438 | 11.481 |
| Network Diameter | 24.000 | 24.000 | 24.000 |
| Average Path Length | 7.934 | 7.948 | 7.948 |

TABLE XII.    ANALYSIS OF NETWORK METRICS AFTER REMOVING NODES WITH HIGHEST BETWEENNESS CENTRALITY

| Sub-community labels | Community 4 | Community 11 | Community 34 |
|---|---|---|---|
| Average Degree | 11.470 | 11.449 | 11.481 |
| Network Diameter | 25.000 | 24.000 | 24.000 |
| Average Path Length | 7.957 | 7.952 | 8.013 |

After removing the node with the highest degree centrality, the average degree across the entire network decreased by 0.072 on average, with a change rate of 6.261‰. The average change in network diameter was 0, and the average path length increased by 0.004 on average, with a change rate of 0.504‰.After removing the node closest to the highest degree centrality, the average degree across the entire network decreased by 0.050 on average, with a change rate of 4.378‰. The average change in network diameter was 0, and the average path length increased by 0.007 on average, with a change rate of 0.924‰ .After removing the node with the highest betweenness centrality, the average degree across the entire network decreased by 0.032 on average, with a change rate of 2.811‰ . The average network diameter increased by 0.333 on average, with a change rate of 13.889‰ , and the average path length increased by 0.038 on average, with a change rate of 4.789‰.

Through the study, it was found that in the directional network derived from the Louvain algorithm, removing the node with the highest degree centrality had the greatest impact on the average degree of the network, while removing the node with the highest betweenness centrality had the greatest impact on the average network diameter. Removing any key node had the greatest impact on the average degree of the network but a smaller impact on the network diameter. Based on these findings, the following recommendations are proposed for the directional shipping network: Key nodes in shipping channels, like port entries and exits, bends in narrow channels, intersections, and areas near breakwaters, are critical for safe and efficient operations. Installing lighthouses, buoys, and navigation systems, providing tugboat services, and implementing Vessel Traffic Services (VTS) ensure vessels navigate safely. Regular dredging at bends, clear navigation marks at intersections, and maintenance of breakwaters enhance safety. Updating electronic charts and preparing emergency plans for avoidance zones further mitigate risks, collectively optimizing the shipping network's safety and efficiency.

## IV. CONCLUSION

Research in the Ningbo-Zhoushan Archipelago utilized AIS data to analyze cargo ship trajectories. Cleaned data identified static and turning points, forming feature regions through clustering. Trajectory patterns were recognized to assess region connectivity, creating a topological map with the Louvain algorithm. Removal of key nodes analyzed their impact, aiding harbor authorities in monitoring high-risk areas and supporting safe navigation and route planning. Limitations include the focus on cargo ships and geographic scope, with future research aiming to include diverse vessels and broader areas, enhancing maritime management insights.

Limitations of the study include the use of AIS data exclusively from cargo ships, resulting in a limited data sample, and the geographical scope confined to the waters of Ningbo-Zhoushan Port. Future research will expand to include different vessel types and broaden the geographical coverage, considering additional factors to enhance the comprehensiveness and applicability of research findings, thereby providing critical insights for optimizing maritime management and enhancing safety.

## ACKNOWLEDGEMENT

This work was supported in part by the National Natural Science Foundation of China (grant nos. 52131101 and 51939001), the Liao Ning Revitalization Talents Program (grant no. XLYC1807046), and the Science and Technology Fund for Distinguished Young Scholars of Dalian (grant no. 2021RJ08).

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
