# OpenReview forum: "Community Detection in Shipping Network  Based on AIS data"
_IEEE.org/ICIST/2024/Conference — IEEE ICIST 2024 Conference Submission_

### Official Review · Reviewer_wTsN · 2024-08-26
**Minor revison**

**Rating:** 7
**Confidence:** 5

**Review:**

This paper focuses on categorizing vessel navigation states into straight sailing, stationary, and turning, with emphasis on stationary and turning as critical features. The writing and language should be improved and polished. After minor revision, it can be accepted as a conference paper.

---

### Official Review · Reviewer_KwkG · 2024-08-29
**Community Detection in Shipping Network Based on AIS data**

**Rating:** 6
**Confidence:** 3

**Review:**

1、The images in the paper are very unclear and the author is advised to replace them.
2、The legends in many of the figures are too small. Examples include figure 2 and figure 4.
3、Multiple Chinese characters appear in the figures?
4、The image in Table 1 is very blurry and so large that it covers the edges of the Table.
5、There are many grammatical errors and formatting issues in the text, and it is recommended that the author carefully check and revise them.

---

### Decision · Program_Chairs · 2024-09-06

Accept (Oral)